# A Lectin AtTL-2 Obtained from *Acropora* aff. *tenuis* Induced Stimualation of Phagocytosis of Symbiodiniaceae

**DOI:** 10.3390/microorganisms13051095

**Published:** 2025-05-08

**Authors:** Mitsuru Jimbo, Nami Kuniya, Yuna Fujimaki, Daiki Yoshikawa, Naoki Kamiya, Haruna Amano, Ko Yasumoto, Ikuko Yuyama, Go Suzuki, Saki Harii

**Affiliations:** 1School of Marine Biosciences, Kitasato University, 1-15-1, Kitasato, Sagamihara 252-0373, Kanagawa, Japana-haruna@kitasato-u.ac.jp (H.A.); yasumoto@kitasato-u.ac.jp (K.Y.); 2Graduate School of Science and Technology for Innovation, Yamaguchi University, 1677-1, Yoshida, Yamaguchi 753-8515, Yamaguchi, Japan; yuyama@yamaguchi-u.ac.jp; 3Fisheries Technology Institute, Japan Fisheries Research and Education Agency, 148, Fukaiohta, Ishigaki 907-0451, Okinawa, Japan; gosuzu@affra.go.jp; 4Tropical Biosphere Research Center, University of the Ryukyus, 3422, Sesoko, Motobu 905-0227, Okinawa, Japan; sharii@cs.u-ryukyu.ac.jp

**Keywords:** coral symbiosis, lectin, phagocytosis, *Acropora* aff. *tenuis*, juvenile polyps

## Abstract

The coral *Acropora* aff. *tenuis* selectively acquired various zooxanthella (Symbiodiniaceae) strains, and one of the selective factors was lectins. The *A*. aff. *tenuis* lectin AtTL-2 was identified as a factor for Symbiodiniaceae acquisition by the coral, but the mechanism is not fully known. The acquisition process involves three steps: chemotaxis, entry into the coral, and phagocytosis. In this study, we examined the function of AtTL-2 in more detail. Immunohistochemistry analysis was performed to examine the distribution of AtTL-2. The effect of AtTL-2 on the number of Symbiodiniaceae acquired was measured in *A.* aff. *tenuis* juvenile polyps with and without *AtTL-2* siRNA treatment. The effect of AtTL-2 fixation was examined by monitoring the acquisition of AtTL-2–fixed beads by *A.* aff. *tenuis*. AtTL-2 was distributed in nematocysts, spirocysts, and around Symbiodiniaceae. AtTL-2 siRNA inhibited the acquisition of Symbiodiniaceae by juvenile polyps. Fixation of AtTL-2 promoted bead acquisition by juvenile polyps more than fixation of bovine serum albumin (BSA). Moreover, more AtTL-2–fixed beads were bound to the Symbiodiniaceae-enclosed cells than BSA-fixed beads. AtTL-2 is released from spirocysts and binds to Symbiodiniaceae. AtTL-2 then promotes the phagocytosis of Symbiodiniaceae by gastrodermal cells of *A.* aff. *tenuis.*

## 1. Introduction

Coral reefs are among the highest biodiversity regions in the world, harboring more than 25% of all organisms [1]. Corals play a significant role in supporting this diversity, as evidenced by reports that coral death leads to the disruption of this ecosystem and to decreases in biodiversity. Corals are classified among cnidaria, and many corals in tropical and subtropical areas symbiose with zooxanthellae, particularly unicellular algae of the family Symbiodiniaceae, which provide carbohydrates that nourish corals [2]. Thus, symbiosis between corals and Symbiodiniaceae is important for coral survival and the maintenance of coral reefs.

Most corals do not harbor symbionts at the fertilized egg stage but acquire them in the larval planulae and juvenile polyp stages, as the photosynthate from Symbiodiniaceae is required for coral growth. Corals selectively acquire Symbiodiniaceae from the environment. Several genera of Symbiodiniaceae have been isolated in sea water surrounding coral reefs, but the corals harbor only a limited number of Symbiodiniaceae genera, including *Cladocopium*, *Durusdinium*, and *Symbiodinium*. Yamashita et al. [3] found that the coral *Acropora* aff. *tenui*s [4] harbors only the minor genera *Symbiodinium*, *Cladocopium*, and *Durusudinium*, whereas *Cladocopium* predominates in the environmental pools around the corals. This discrepancy suggests that corals select specific Symbiodiniaceae for acquisition through attraction and phagocytosis using as yet unknown factors.

Light is one factor that attracts Symbiodiniaceae. Green light was shown to attract some Symbiodiniaceae [5], such as those isolated from the giant clam, *Tridacna crocea*. By contrast, *S. microadriaticum*, which is taken up by *A.* aff. *tenuis*, is attracted to blue light rather than green light, whereas other genera of Symbiodiniaceae are not [6]. The wavelength of the blue light was similar to that of the fluorescence of planula larvae rather than green fluorescent protein in the polyps. It is plausible that blue light attracts some Symbiodiniaceae genera.

Various chemicals also attract symbionts. Fitt [7] reported that *Symbiodinium microadriaticum* is attracted to nitrogen derivatives under starvation conditions. Trehalose released from Symbiodiniaceae attracts planulae larva of *Fungia scutaria* [8]. Lectins, which are carbohydrate-binding proteins, also function as attractants. Acquisition of Symbiodiniaceae from the environment by the sea anemone *Aiptasia pallida* is inhibited by glycosidase treatment [9]. Acquisition of Symbiodiniaceae from the environment by the coral *Fungia scutaria* is also inhibited by glycosidase treatment or the addition of the mannose-binding lectin concanavalin A, suggesting that corals acquire symbionts using carbohydrates [10]. Carbohydrates with varied structures cover all cells of all organisms as components of either the cell wall or extracellular matrix. Lectins that bind specific carbohydrates mediate selective binding to pathogens to trigger an innate immune response, which serves as the primary immune system in invertebrates, including cnidaria. Symbiodiniaceae, as different species, can elicit immune responses in some corals [11]. Thus, it is plausible that a lectin is involved in the acquisition of Symbiodiniaceae.

The easily identifiable hard coral *Acropora* aff. *tenuis* is found around Okinawa, Japan. The larvae of *A.* aff. *tenuis* can be kept for several months without undergoing metamorphosis. Moreover, the planula larvae can be artificially induced to metamorphose by the addition of the neural peptide Hym-248 [12], and the resulting juvenile polyps can acquire Symbiodiniaceae strains [13,14]. The coral *A.* aff. *tenuis* was selected to investigate the acquisition of Symbiodiniaceae. Juvenile polyps of *A.* aff. *tenuis* acquired *Symbiodinium microadriaticum* NBRC102920, and this acquisition was inhibited in the presence of galactose, *N*-acetyl-D-galactosamine (GalNAc), and *N*-acetyl-D-glucosamine (GlcNAc). The lectin ActL, which binds to GalNAc and GlcNAc, was shown to attract some Symbiodiniaceae strains, and those that were attracted in greatest abundance were acquired by juvenile polyps [15].

Another lectin, *A.* aff. *tenuis* tachylectin-2 (AtTL-2), was shown to bind only to GalNAc. The acquisition of Symbiodiniaceae was inhibited by an anti–AtTL-2 antibody [14]. However, AtTL-2 did not attract Symbiodiniaceae in a study using a capillary assay [16]; thus, the function of AtTL-2 remains unknown. Symbiodiniaceae reside within the cells of host corals, and the acquisition process involves three steps: (1) attraction of Symbiodiniaceae to polyps, (2) entry of Symbiodiniaceae into the gastrovascular cavity, and (3) entry of Symbiodiniaceae into gastrodermal cells. Fitt et al. (1983) reported that Symbiodiniaceae enter host cells via endocytosis (phagocytosis) [17]. Phagocytosis is a process of the immune response, and lectins bind to pathogens and promote their phagocytosis. As this immunological event is similar to the third step of the Symbiodiniaceae acquisition process, in this study, we examined whether the lectin AtTL-2 promotes the phagocytosis of Symbiodiniaceae by *A.* aff. *tenuis.*

## 2. Materials and Methods

### 2.1. Materials

The anti–AtTL-2 antibody was obtained from Professor Shunichiro Kawabata, Kyushu University (the antibody was raised in rabbits using AtTL-2 as the antigen). Planulae were obtained from *A.* aff. *tenuis* corals around Ishigaki Island or Sesoko Island, Okinawa Prefecture, Japan. Bundles of *A.* aff. *tenuis* corals spawned by the addition of hydrogen peroxide [18,19] were collected with pipettes, mixed to promote fertilization and then reared in 0.2 µM filtered seawater. After transfer to the laboratory, the obtained larvae were kept in Marine Art SF-1 artificial seawater (ASW: Osaka Yakken, Osaka, Japan) supplemented with 5 µg/mL ampicillin and 2.5 µg/mL kanamycin. *Symbiodinium microadriaticum* strain NBRC102920 was purchased from NITE (Tokyo, Japan). NBRC102920 was cultured in Daigo’s IMK medium (Fuji Film Wako Pure Chemical, Osaka, Japan) and incubated at 27 °C in 40 µMol photon m^−2^ s^−1^ under a 12 h:12 h light:dark cycle.

### 2.2. RNA Interference

RNA interference (RNAi) analyses were performed using HiPerFect transfection reagent (QIAGEN, Venlo, The Netherlands) according to the manufacturer’s protocol. Table 1 lists the siRNAs used in the study. To examine RNAi of green fluorescent protein (GFP), 10 planulae in 200 µL of filtered ASW (FASW) were incubated with 2 µM hydra neural peptide Hym-248 in 8 wells of a Lab-Teck II chambered coverglass (Thermo Fisher Scientific, Waltham, MA, USA) for 24 h to allow for polyp metamorphosis. A total of 190 µL of FASW was then removed and added to each well and allowed to stand for 2 h. Next, 50 µL of 220, 660, or 2000 nM GFP siRNA or random siRNA (Table 1) was mixed with 5 µL of HiPerFect transfection reagent and allowed to stand for 10 min, and then 10 µL of this mixture was added to the polyps. After 24 h, the expression of GFP or red fluorescent protein (RFP) by polyps was examined using an LSM-800 confocal laser scanning microscope (Carl Zeiss, Oberkochen, Germany). The resulting images were analysed using Fiji software (version 2.16.0/1.54p) [20] and projected onto the Z axis according to maximum intensity; the mean GFP and RFP fluorescence was then determined.

To examine RNAi of AtTL-2 expression, 400 nM AtTL-2 siRNA was added instead of GFP siRNA. At 24 h after siRNA treatment, 400 µL of ASW was replaced with 400 µL of fresh ASW and incubated for 2 h, after which 5000 Symbiodinium microadriaticum NBRC102920 cells were added and incubated at 28 °C with 20 µMol photon m^−2^ s^−1^. After 24 h, GFP and in vivo chlorophyll a fluorescence were examined using an LSM-800 microscope. The resulting images were analyzed using Fiji software (version 2.16.0/1.54p). The images were projected onto the Z axis according to maximum intensity, and then in vivo chlorophyll a fluorescence of NBRC102920 was selected and measured. This experiment was carried out several times over a period of three years. The representative result was used.

### 2.3. Immunohistochemistry

Anti–AtTL-2 antibody was labeled with DyLight 488 (Thermo Fisher Scientific) according to the manufacturer’s protocol. Small coral fragments were fixed overnight in Buin’s solution at 4 °C, and then the solution was exchanged with 70% ethanol. Polyps were dehydrated in 80%, 90%, and 95% absolute ethanol for 1 h each, followed by xylene for 1 h, and then embedded in paraffin. The tissue was then sectioned at 5 µM thickness using a rotary microtome Leica RM2125 RTS (Leica Biosystems, Nussloch, Germany) and placed on MAS-GP coated slide glass (Matsunami Glass, Osaka, Japan). The paraffin was removed by two treatments with xylene, and the samples were rehydrated by sequential treatment with 95%, 90%, 80%, and 70% ethanol for 2 min each. After surrounding the sections with ImmEdge PEN (Vector Laboratories, Newark, CA, USA), the sections were blocked with 2% normal rabbit serum in PBS for 1 h at room temperature. After removing the blocking solution, DyLight488-labeled anti–AtTL-2 antibody in Can Get Signal Immunostain B (TOYOBO, Osaka, Japan) was added to each section and incubated for 1 h in the dark. After washing with PBS three times, the sections were embedded using Fluoromount/Plus (Diagnostic BioSystems, Pleasanton, CA, USA). Immunofluorescence was observed using an IX71 fluorescence microscope (Evident, Tokyo, Japan) with blue excitation.

### 2.4. Preparation of AtTL-2–Fixed Beads

AtTL-2 was prepared using pET16b according to Kuniya et al. [14]. Recombinant AtTL-2 (rAtTL-2) protein contained a histidine-tag at the N-terminus of the mature AtTL-2 sequence. rAtTL-2 was purified using a HiTrap chelating column according to the manufacturer’s protocol. The buffer of purified rAtTL-2 was changed to 100 mM MES (pH 5.2) using VIVASPIN 500 MWCO 10k spin columns (Sartorius, Göttingen, Germany), and the concentration was adjusted to 1 mg/mL.

To prepare labeled beads, Fluoresbrite 641 carboxylate microspheres (1.75 µM; Polysciences, Warrington, PA, USA) were used. Vivaspin 500 MWCO 300k spin columns (Sartorius) were washed with water, and then 50 µL of beads was added and centrifuged at 1000× g for 4 min at room temperature. After removal of the filtrate, the beads were washed three times with 400 µL of 100 mM MES (pH 5.2). The washed beads were resuspended in 170 µL of 100 mM MES buffer (pH 5.2) containing 200 mg/mL water-soluble carbodiimide (Dojindo Laboratories, Kumamoto, Japan) and mixed for 15 min at room temperature. After adjusting the volume to 200 µL with 100 mM MES buffer (pH 5.2), 200 µL of rAtTL-2 was added and mixed for 1 h. The samples were then centrifuged at 1000× *g* for 3 min, and the filtrate was removed. A total of 200 µL of FASW was added to the residual beads and centrifuged at 1000× *g* for 3 min twice. After addition of 200 µL of FASW, the beads were resuspended and transferred to a new tube. As a negative control, bovine serum albumin (BSA) was used instead of rAtTL-2. The density of beads was measured using 10-fold diluted bead samples on a slide glass using an LSM-800 laser scanning microscope with red fluorescence (Ex 640 nm, Em 656–700 nm).

### 2.5. Evaluation of Phagocytosis of AtTL-2–Fixed Beads

Eight planulae were metamorphosed for 72 h in a well of a Lab-Tek II chambered coverglass using 2 µM Hym-248 containing FASW to let polyps to the stage that acquire the most Symbiodiniaceae [12]. After changing FASW with fresh FASW, the larvae were kept at 25 °C for 1 h, and then AtTL-2–labeled or BSA-labeled beads were added to a final concentration of 1.0 × 10^5^ particles/mL and incubated for 6 h. Each polyp was imaged using an LSM-800 confocal laser scanning microscope using Z-slices. The resulting images were analyzed using Fiji [20] and processed using max-intensity Z projection. The bead area was selected (greater than threshold) using red fluorescence (Ex 488 nm, Em 656–700 nm), and the total area inside the polyps was measured. These area values were compared using the Mann–Whitney test with GraphPad Prism software, ver. 10.4.0. This experiment was carried out 5 times over two years and the representative result was shown.

### 2.6. Binding of AtTL-2–Fixed Beads to Polyp Cells

The center of a polyp that had acquired Symbiodiniaceae strain AJIS2-C2 was pricked with a toothpick and then mixed with 200 µL of Accutase cell detachment solution (Innovative Cell Technologies, San Diego, CA, USA) containing 0.5 M NaCl and incubated for 30 min. After mixing, the solution was transferred to a 1.5 mL tube. Residual cells were collected using 100 µL of Ca^2+^-free seawater (391.1 mM NaCl, 10.2 mM KCl, 15.7 mM MgSO_4_, 51.4 mM MgCl_2_, 21.1 mM Na_2_SO_4_, 3.0 mM NaHCO_3_) and collected in the same tube. The mixture was centrifuged at 3000× *g* at room temperature for 10 min. After removal of the supernatant, the pellet was washed with Ca^2+^-free seawater.

The cells were mixed with 1.0 × 10^6^ particles/mL of AtTL-2–fixed beads and incubated at room temperature for 1 h. After centrifugation at 3000× *g* for 10 min, the resulting pellet was resuspended in 2 µL of Ca^2+^-free seawater and 1 µL of NucleoSeeing nucleus stain solution (Funakoshi Co., Ltd., Tokyo, Japan) and incubated for 15 min at room temperature in the dark. Next, 2 µL of the solution was placed on an MAS-GP type A slide glass (Matsunami Glass), covered with a cover glass, and observed using an LSM-800 confocal laser scanning microscope (Ex 488 nm and 640 nm, Em 656–700 nm). The resulting images were processed using max-intensity Z projection. The area of chlorophyll *a* fluorescence (Ex 488 nm) or chlorophyll *a* and beads fluorescence (Ex 640 nm) was selected, and then the circularity, an indicator of distortion, was measured. As a negative control, cells treated with BSA-fixed beads or intact cells were observed. This experiment was carried out three times, and the representative result was used.

### 2.7. Statistical Analysis

All statistical analyses were performed using GraphPad Prism software (v. 10.4.0).

## 3. Results

### 3.1. Distribution of AtTL-2

Sections of tentacle were stained with anti–AtTL-2 antibody (Figure 1). Staining was observed around Symbiodiniaceae and within spirocysts, which are a type of cnidocyst.

### 3.2. RNAi Analysis of AtTL-2

RNAi is an effective tool for repressing gene expression in many organisms, including corals [21]. As shown in Figure 2a, the addition of GFP siRNA at concentrations in the range 11–100 nM reduced GFP fluorescence but did not affect RFP fluorescence. These data confirmed that siRNA could be utilized in the present study.

AtTL-2 siRNA was added to polyps to examine the effect on the acquisition of Symbiodiniaceae (Figure 2b). Polyps treated with AtTL-2 siRNA acquired significantly fewer Symbiodiniaceae NBRC102920 cells than untreated polyps, suggesting that AtTL-2 is involved in the acquisition of Symbiodiniaceae.

### 3.3. Acquisition of AtTL-2–Fixed Beads by Juvenile Polyps

To examine the effects of AtTL-2 on the acquisition of Symbiodiniaceae, juvenile polyps were mixed with AtTL-2–fixed beads. Polyps mixed with AtTL-2–fixed beads exhibited red fluorescence, but polyps mixed with BSA-fixed beads exhibited no fluorescence (Figure 3a,b). The area of beads within the polyps (red fluorescence area) was significantly greater in juvenile polyps treated with AtTL-2–fixed beads than that of polyps treated with BSA-fixed beads.

### 3.4. Binding of AtTL-2–Fixed Beads to Symbiodiniaceae-Containing Cells

Symbiodiniaceae cells were encapsuled by host gastrodermal cells. If AtTL-2 promotes the phagocytosis of Symbiodiniaceae cells, the AtTL-2–fixed beads would bind to gastrodermal cells. To examine this hypothesis, the binding of AtTL-2–fixed beads to gastrodermal cells was evaluated. Images obtained at an excitation wavelength of 640 nm enabled simultaneous in vivo detection of chlorophyll *a* fluorescence of Symbiodiniaceae and the fluorescence of beads, whereas images obtained at an excitation wavelength of 488 nm would exhibit only in vivo chlorophyll *a* fluorescence. Thus, the binding of beads was examined based on circularity in images obtained at an excitation wavelength of 640 nm, as the binding of beads distorts the shape of Symbiodiniaceae-containing cells. When AtTL-2–fixed beads were added to coral cells, the circularity of these cells was lower than that observed at Em 488 nm, whereas the addition of BSA-fixed beads resulted in similar circularity at both excitation wavelengths of 488 nm and 640 nm (Figure 4). These results indicated that AtTL-2–fixed beads were bound to gastrodermal cells.

## 4. Discussion

AtTL-2 was distributed in spirocysts, a type of cnidocyst (Figure 1) [22]. Spirocysts release adhesive substances to attract prey. Cnidocysts sometimes release these substances without stimulation. In *Exaiptasia diaphana*, for example, the proportion of spirocysts is lower in the symbiotic stage than the aposymbiotic stage [23]. Thus, spirocysts could release to either attract prey or obtain Symbiodiniaceae.

Polyps treated with AtTL-2 siRNA acquired fewer Symbiodiniaceae (Figure 2). Treatment with anti–AtTL-2 antibody reduced the number of Symbiodiniaceae acquired by polyps [14]. These results confirmed that AtTL-2 plays a role in the acquisition of Symbiodiniaceae by polyps.

AtTL-2 appears to be involved in the acquisition of Symbiodiniaceae by *A.* aff. *tenuis*, as some lectins have been identified on the surface of Symbiodiaceae cells within host cells, and inhibitors of lectin binding have been shown to prevent the acquisition of Symbiodiniaceae [14,24,25]. To examine the role of AtTL-2 in this regard, we first determined the distribution of AtTL-2 in polyps (Figure 1). An anti–AtTL-2 antibody stained some Symbiodiniaceae and spirocysts, a type of cnidocyst, suggesting that spirocysts are important for the capture of prey, as they are adhesive and their abundance is elevated in aposymbiotic starved states [23,26]. We found that polyps sometimes released cnidocysts without stimulation. Moreover, Kuniya et al. (2015) reported that anti–AtTL-2 antibodies inhibited the acquisition of Symbiodiniaceae NBRC102920 by juvenile polyps [14]. Our data and those of previous studies thus suggest that AtTL-2 is released from spirocysts to the external environment, where it binds to the surface of Symbiodiniaceae cells. Host gastrodermal cells then phagocytose the Symbiodiniaceae. The surface proteins of Symbiodiniaceae cells then play a role in the symbiotic relationship with *A.* aff. *tenuis*.

To evaluate the role of AtTL-2 in the acquisition of Symbiodiniaceae in detail, juvenile polyps were treated with AtTL-2 siRNA (Figure 2). The number of Symbiodiniaceae acquired by polyps was reduced by treatment with AtTL-2 siRNA. As both the anti–AtTL-2 antibody and N-acetyl D-galactosamine, which binds to AtTL-2, inhibited the acquisition of Symbiodiniaceae by juvenile polyps [14], these results supported the hypothesis that AtTL-2 plays an important role in the acquisition of Symbiodiniaceae by *A.* aff. *tenuis* polyps.

The process of Symbiodiniaceae acquisition involves three steps: (1) attraction of Symbiodiniaceae via chemotaxis/phototaxis [5,6,8,15,16], (2) entry into polyps, and (3) phagocytosis of Symbiodiniaceae by gastrodermal cells [17]. A previous study found that AtTL-2 does not attract Symbiodiniaceae NBRC102920 (Takeuchi, personal communication). Since the water current from the pharynx to the gastrovascular cavity enlarges polyps [27], step (2) (entry into polyps) could utilize this current. Thus, AtTL-2 appears to be involved in step (3), as some lectins have been shown to promote phagocytosis by macrophage through binding to pathogens [28]. The promotion of phagocytosis was examined in the present study using AtTL-2–fixed beads (Figure 3). Juvenile polyps acquired AtTL-2–fixed beads more readily than BSA-fixed beads. The AtTL-2–fixed beads bound to Symbiodiniaceae-containing cells to a greater extent than did BSA-fixed beads (Figure 4). These results indicate that the lectin AtTL-2 binds to a receptor on the surface of gastrodermal cells. Lectins play important roles in invertebrates, which have only an innate immune system for defense against pathogens, and lectins are primary recognition molecules for pathogens. The lectin AtTL-2 is similar to tachylectin-2 (TL-2) purified from the hemolymph of Tachytus tridentatus. In previous studies, TL-2 agglutinated bacteria [29,30]. Moreover, in T. tridentatus, innate immunity-related components such as C3 were shown to induce recognition of pathogens, and the lectin TL-5A was involved in the activation of these components [31]. These previous studies speculated that TL-2 is also involved in the activation of C3. However, some cnidarians also express innate immunity components such as C3, MASP, and Bf [32,33,34]. These components were found to be distributed in gastrodermal cells, which harbor resident Symbiodiniaceae. Data indicating that AtTL-2 promotes the acquisition of beads and that cells with bound AtTL-2 enclose zooxanthella suggest that AtTL-2 plays a role in promoting the acquisition of Symbiodiniaceae by binding to one or more innate immunity components.

## 5. Conclusions

In this paper, we showed that AtTL-2 stimulates coral’s phagocytosis of Symbiodiniaceae, and AtTL-2 could help bleached corals to survive. Due to climatic variations, coral bleaching has become a major worldwide problem. This bleaching results from the loss of Symbiodiniaceae by coral cells, which leads to prolonged stress that can in turn lead to mortality of the coral. This phenomenon could be due to the absence of symbiotic Symbiodinium or to a reduction in the ability of the corals to acquire Symbiodiniaceae. Previous studies have indicated that corals acquire a limited number of Symbiodiniaceae strains. Thus, in subsequent studies, we plan to examine how AtTL-2 promotes the acquisition of Symbiodiniaceae in greater detail.

## Figures and Tables

**Figure 1 microorganisms-13-01095-f001:**
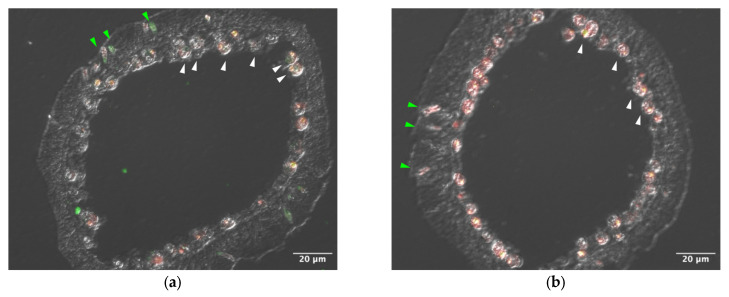
Distribution of AtTL-2 in tentacles. Sections of tentacles were stained with anti–AtTL-2 antibody (**a**) or treated without the antibody (**b**) and observed using an LSM-510 confocal laser scanning microscope (Zeiss). Red and green indicate fluorescence of chlorophyll a and anti–AtTL-2 antibody, respectively. Green and white arrowheads indicate spirocysts and Symbiodiniaceae, respectively.

**Figure 2 microorganisms-13-01095-f002:**
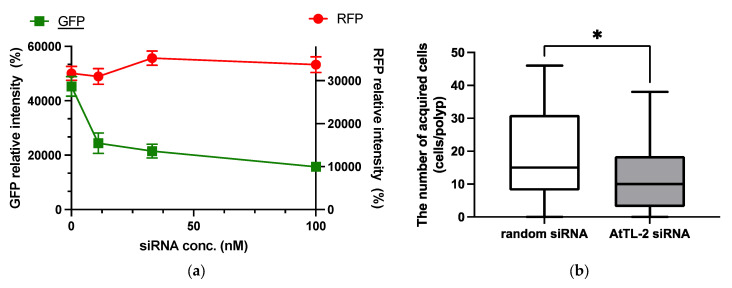
Effect of GFP siRNA and AtTL-2 siRNA on *Acropora* aff. *tennis* polyps. (**a**) After treatment with GFP siRNA, planulae were allowed to metamorphose and kept in FASW. After 24 h, the metamorphosed larvae were observed using an LSM-800 laser confocal microscope. The X- and Y-axis show siRNA concentration and GFP or RFP intensity, respectively. Red and green lines indicate the fluorescence intensity of RFP and GFP, respectively. (**b**) One day after metamorphosis of juvenile polyps, AtTL-2 siRNA or random siRNA was added. The polyps were incubated with Symbiodiniaceae NBRC102920 for 24 h. The number of NBRC102920 taken up by juvenile polyps was then determined using an LSM-800 confocal laser microscope. * *p* < 0.05 (Mann–Whitney test, *n* = 23).

**Figure 3 microorganisms-13-01095-f003:**
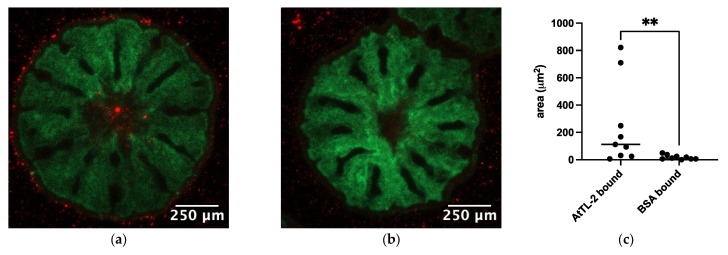
Acquisition of AtTL-2–fixed beads by juvenile polyps. Juvenile polyps were mixed with 1.5 × 10^5^ particles/mL of beads fixed with AtTL-2 (**a**) or BSA (**b**) and incubated for 24 h. The polyps were then observed using an LSM-800 confocal laser scanning microscope. Green and red indicate the fluorescence of GFP and beads, respectively. (**c**) The area of beads within the polyps was measured using Fiji image analysis software, and the data were analyzed using the Mann–Whitney test with GraphPad Prism software, ver. 10.4.0. ** *p* < 0.01.

**Figure 4 microorganisms-13-01095-f004:**
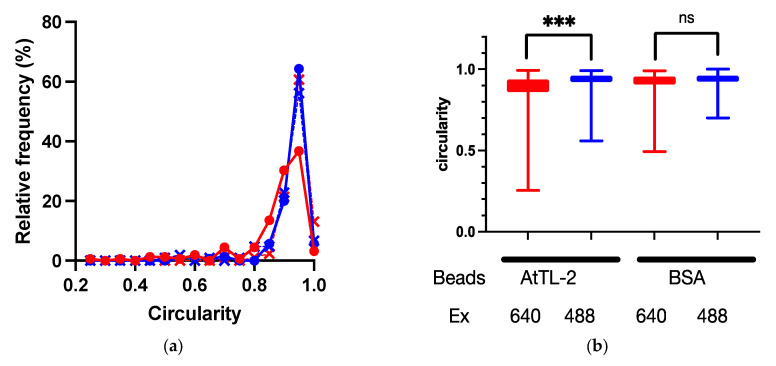
Binding of AtTL-2–fixed beads to polyp cells. Cells separated from polyps were mixed with AtTL-2–fixed or BSA-fixed beads and allowed to stand for 1 h. The mixture was then observed using an LSM-800 confocal laser scanning microscope at an excitation wavelength of 488 nm or 640 nm. The area occupied by Symbiodiniaceae or Symbiodiniaceae and beads was selected, and the circularity of each area was measured. Red and blue lines indicate observations at excitation wavelengths of 640 nm and 488 nm, respectively. (**a**) Distribution of circularity of area of Symbiodiniaceae with beads. Circles indicate the circularity of polyp cells with AtTL-2–fixed beads, respectively. Crosses indicate the circularity of polyp cells with BSA-fixed beads. (**b**) Circularity of Symbiodiniaceae only and Symbiodiniaceae + beads indicated using box and whisker plots. AtTL-2 and BSA indicate AtTL-2–fixed and BSA-fixed beads, respectively. Ex indicates excitation wavelength. Data were compared using the Kruskal–Wallis test with multiple comparisons. *** *p* < 0.001; ns, not significant.

**Table 1 microorganisms-13-01095-t001:** siRNAs used for RNAi analyses.

siRNA Name	Target mRNA (Accession No.)	Sequences
AtTL-2	Attl-2 (AB972924)	CCGCAUAUGCGAGUGACAAdTdT UUGUCACUCGCAUAUGCGGdTdT
GFP	Green fluorescent protein (AB626608)	GAGGUGAUCUGGCUAUGUUdTdT AACAUAGCCAGAUCACCUCdTdT
random	-	AAUUGGGUCUGUUGAGGCUdTdT AGCCUCAACAGACCCAAUUdTdT

## Data Availability

The original contributions presented in this study are included in the article. Further inquiries can be directed to the corresponding author.

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
