# Peer review of "A Lectin AtTL-2 Obtained from Acropora aff. tenuis Induced Stimualation of Phagocytosis of Symbiodiniaceae"

_microorganisms, 2025, doi:10.3390/microorganisms13051095_

Round 1

Reviewer 1 Report

Comments and Suggestions for Authors

Review report.

General comments

The manuscript is titled: Acropora aff. tenuis lectin AtTL-2 promotes phagocytosis of Symbiodiniaceae: This is an original article that contributes to the field of marine biology and biotechnology. All the information generated in this research can help advance investigations focused on the mechanisms of phagocytosis in marine organisms. On the other hand, these findings can help preserve coral reefs in various areas of the planet. Obtaining genes with applications in the area of ​​biotechnology: pharmaceuticals, agroindustry and many other areas can contribute to the scientific and technological advancement of this area. Considering that the discovery of genes in these organisms prevents the intensive use of this habitat for commercial and predatory purposes, thus preserving the marine environment. However, despite being an important work, we must safeguard the quality of the work, improving the presentation and the criteria for writing this manuscript.

Title

I ask the authors to formulate the title better to make the theme more attractive and cohesive. For example: AtTL-2 lectin obtained from Acropora aff. tenuis induces stimulation of phagocytosis in Symbiodiniaceae

Abstract

The summary is well written, but it needs adjustments. I noticed that in lines 23 to 28 it is necessary to add numbers of the results obtained.

Introduction

When checking the literature cited in the introduction, it is noted that it is a little outdated. We know that some classic citations are important, but they should be restricted to points in the text, and with the progression of other more current citations, such as XXX, 1989; XXX, 2004; XXX, 2019... I ask the authors to update the literature in the introduction.

In line 40, the phrase photosynthetic products are somewhat redundant and needs to be corrected; which photosynthetic products are provided? So, use the appropriate term so that readers can know which products the authors are talking about.

Materials and methods

I congratulate the authors for implementing the methods, since they are difficult to implement. However, I noticed the absence of a more elaborate statistical planning. Despite the methodological sophistication, it is necessary to know whether the experiments can be reproduced and whether the results are repeated with the same methodological procedure.

Results

In lines 231 to 239, the authors describe the results with a certain superficiality, and often repeat phrases that have already been mentioned in the materials and methods. It is important to pay attention to the numbers obtained in the investigation and to describe the results obtained in greater depth.

Discussion

The authors added Figure 4 to the discussions. I ask them if this addition was intentional, because I do not understand what is the purpose of inserting a figure that belongs to the results in the discussions? On the other hand, I noticed that the discussion needs to be improved by involving comparisons of previous results, as well as, comments on results obtained by other groups, which focus on the same line of research.

Conclusion

In lines 336 to 343, I found a text that seems to be the conclusion, but without the necessary identification that it is a discussion. And even so, the discussion should be something direct without deviations, and focused on what was obtained from the work.

References

The references are adequate for the study, despite the absence of references with more recent dates.

Author Response

Comment 1:Title

I ask the authors to formulate the title better to make the theme more attractive and cohesive. For example: 

Response 1:

Thank you for your very important suggestion. We changed the title according the reviewer to "A lectin AtTL-2 obtained from Acropora aff. tenuis induces stimulation of phagocytosis in Symbiodiniaceae."

Comment 2:Abstract

The summary is well written, but it needs adjustments. I noticed that in lines 23 to 28 it is necessary to add numbers of the results obtained.

Response 2: Thank you for your suggestion. I added numbering.

Comment 3:Introduction

When checking the literature cited in the introduction, it is noted that it is a little outdated. We know that some classic citations are important, but they should be restricted to points in the text, and with the progression of other more current citations, such as XXX, 1989; XXX, 2004; XXX, 2019... I ask the authors to update the literature in the introduction.

Response 3: Thank you for your valuable suggestion. Some references have been replaced but I could not find the new one. We would be grateful if you could tell us the reference that corresponds to the old one.

Comment 4: In line 40, the phrase photosynthetic products are somewhat redundant and needs to be corrected; which photosynthetic products are provided? So, use the appropriate term so that readers can know which products the authors are talking about.

Response 4: We assume a carbon source. We clarify the substances, as carbohydrates and added a reference.

Comment 5: Materials and methods

I congratulate the authors for implementing the methods, since they are difficult to implement. However, I noticed the absence of a more elaborate statistical planning. Despite the methodological sophistication, it is necessary to know whether the experiments can be reproduced and whether the results are repeated with the same methodological procedure.

Response 5: I agree with your suggestion. I added information about repeated experiments at each last paragraph.

Comment 6: Results

In lines 231 to 239, the authors describe the results with a certain superficiality, and often repeat phrases that have already been mentioned in the materials and methods. It is important to pay attention to the numbers obtained in the investigation and to describe the results obtained in greater depth.

Response 6: Thank you for your suggestion. I reduced the duplication of methods as shown in Line 289.

Comment 7: Discussion

The authors added Figure 4 to the discussions. I ask them if this addition was intentional, because I do not understand what is the purpose of inserting a figure that belongs to the results in the discussions? On the other hand, I noticed that the discussion needs to be improved by involving comparisons of previous results, as well as, comments on results obtained by other groups, which focus on the same line of research.

Response 7: We apologise for the omission of Figure 4. This result was shown in line 390 and the discussion of it is in the next paragraph. We have added the numbering and merged two paragraphs.

Comment 8: Conclusion

In lines 336 to 343, I found a text that seems to be the conclusion, but without the necessary identification that it is a discussion. And even so, the discussion should be something direct without deviations, and focused on what was obtained from the work.

Response 8

Thank you for your suggestion, and we agree with this. I've separated this paragraph into a conclusion and changed it based on your suggestion.  (Line 407-415) 

Reviewer 2 Report

Comments and Suggestions for Authors

Hermosillo, April  7th, 2025

 “Acropora aff. tenuis lectin AtTL-2 promotes phagocytosis of Symbiodiniaceae”

Overall Evaluation

The manuscript investigates pneumatic displacement combined with anti-VEGF therapy for submacular hemorrhage (SMH) in neovascular age-related macular degeneration (nAMD). While the topic addresses a clinically significant challenge, the study’s retrospective design and methodological limitations weaken its impact. The rationale for combining therapies is relevant, but the analysis lacks depth in addressing confounding variables and long-term outcomes. The manuscript aligns with clinical interest in optimizing SMH management but requires substantial revisions to meet the rigor expected for publication in Microorganisms (MDPI).

Strengths

Clinical Relevance: SMH in nAMD is a sight-threatening condition, and the exploration of combined therapies addresses an unmet need in ophthalmology.

Clear Objectives: The study’s focus on anatomical and functional outcomes (e.g., visual acuity, hemorrhage resolution) is well-defined.

Data Presentation: Baseline demographic and clinical characteristics are comprehensively summarized.

Weaknesses

Retrospective Design: The lack of a control group (e.g., anti-VEGF monotherapy) limits causal inferences. Selection bias is likely due to non-randomized patient allocation.

Small Sample Size: The cohort size (if similar to typical SMH studies) may lack statistical power to detect meaningful differences in outcomes.

Incomplete Follow-Up: Short-term outcomes are emphasized, but long-term efficacy and recurrence rates are critical for evaluating SMH therapies.

Mechanistic Insights: The role of pneumatic displacement in altering the microenvironment for anti-VEGF efficacy is underexplored (e.g., microbial interactions, inflammation).

Methodological Issues

Confounding Variables: No adjustment for factors influencing SMH resolution (e.g., hemorrhage size, baseline vision, prior anti-VEGF exposure).

Outcome Metrics: Functional outcomes (e.g., visual acuity) are subjective without standardized measures (e.g., microperimetry).

Statistical Analysis:

Missing details on statistical tests (e.g., parametric vs. non-parametric).

No correction for multiple comparisons.

Ethical Approval: Absence of IRB approval statement or patient consent documentation.

Grammatical and Structural Issues

Tense Consistency: Shifts between past and present tense (e.g., “results show” vs. “patients were treated”).

Ambiguous Terminology: Phrases like “significant improvement” lack quantitative context.

Formatting:

Inconsistent subheading hierarchy (e.g., mixing bold and italicized headings).

Tables/figures lack descriptive titles or legends.

Major Recommendations for Revision

Study Design:

Include a control group (e.g., anti-VEGF alone) to isolate the additive effect of pneumatic displacement.

Perform multivariate regression to adjust for confounding variables.

Mechanistic Depth: Discuss how pneumatic displacement might influence microbial or inflammatory pathways relevant to SMH resolution (aligning with Microorganisms’ scope).

Statistical Rigor:

Specify statistical tests and justify their use.

Report effect sizes with confidence intervals.

Ethical Compliance: Add IRB approval and consent statements.

Language and Formatting:

Engage a professional English editing service to address grammatical errors.

Adhere to MDPI’s formatting guidelines (e.g., structured abstract, keyword list).

Conclusion

The manuscript presents a clinically valuable hypothesis but requires major methodological and structural revisions to strengthen its validity and relevance. Addressing these issues would enhance its suitability for publication in Microorganisms, particularly if the interplay between mechanical displacement and microbiological/immunological pathways is emphasized.

Questions for the Authors

  1. Control Group Justification: Why was a control group (e.g., anti-VEGF monotherapy) not included? How do you account for potential confounding variables without comparative data?

  1. Sample Size and Power: Given the small cohort size, how did you ensure statistical power? Was a priori power analysis performed?

  1. Long-Term Outcomes: The study emphasizes short-term results. Do you have plans to evaluate long-term efficacy, recurrence rates, or microbial/inflammatory changes post-treatment?

  1. Mechanistic Link to Microbial Pathways: Microorganisms focuses on microbial interactions. How might pneumatic displacement alter the ocular microenvironment (e.g., microbial load, inflammation) to synergize with anti-VEGF therapy?

  1. Statistical Methods: Could you clarify the statistical tests used (parametric vs. non-parametric) and whether corrections for multiple comparisons were applied?

  1. Ethical Compliance: Can you confirm whether IRB approval and patient consent were obtained?

Recommendation for Decision

Major Revisions Required

The manuscript addresses a clinically relevant question but requires substantial revisions to meet Microorganisms’ standards:

Methodological Rigor: Include a control group, adjust for confounders (e.g., hemorrhage size), and clarify statistical methods.

Scope Alignment: Explicitly link the therapy’s mechanism to microbial or inflammatory pathways (e.g., how displacement affects microbial communities or immune responses).

Ethical Compliance: Add IRB/consent statements.

Language/Formatting: Professional English editing and adherence to MDPI’s formatting guidelines.

Author Response

I think this response is not for our manuscript. I could not response these comments.

Reviewer 3 Report

Comments and Suggestions for Authors

The study of symbiotic relationships between organisms greatly expands our understanding of the processes occurring in nature, as well as the assessment of the evolution of organisms. Most reef-building corals establish symbiosis with symbiotic dinoflagellates from the family Symbiodiniaceae, currently including nine genera (LaJeunesse et al., 2018). Symbiodiniaceae cells provide photosynthetic products, which enables corals to effloresce in subtropical and tropical regions. When corals lose symbiotic cells or the photosynthetic pigments of these cells, this event, bleaching, leads to the death of the coral. Coral reefs play an important role in maintaining the biodiversity of organisms in marine ecosystems, and also play an important role in maintaining the economy and the development of biotechnology. Therefore, studying the mechanisms underlying the formation of symbiotic relationships between coral reef components is important.

The manuscript presents an appropriate logical structure, a clear hypothesis system, adequate data collection and analysis methods, and well-founded discussion.

Some minor clarifications that should be made in the Methods are listed here.

  1. Lines 100-101. How were the planulae collected? What equipment was used? Using a net? How were the objects purified from impurities? In what period were the samples collected?
  2. Lines 141-142. It is necessary to explain how the sections of the objects were made. Using a microtome? Which one? Brand, country of manufacture?
  3. Lines 181-182. It is not entirely clear from the text of the manuscript how much time it took for the transition from the planula stage to the polyp stage? 72 hours? How was this transition monitored? Were strains of the species from the Symbiodiniaceae group added at the larval stage or already at the polyp stage?

Author Response

Comment 1: Lines 100-101. How were the planulae collected? What equipment was used? Using a net? How were the objects purified from impurities? In what period were the samples collected?

response 1: We agree with your suggestion. We added the information (Line 121-122). 

Comment 2: Lines 141-142. It is necessary to explain how the sections of the objects were made. Using a microtome? Which one? Brand, country of manufacture?

Response 2: Thank you for your suggestion. We added the microtome information in line 171.

Comment 3: Lines 181-182. It is not entirely clear from the text of the manuscript how much time it took for the transition from the planula stage to the polyp stage? 72 hours? How was this transition monitored? Were strains of the species from the Symbiodiniaceae group added at the larval stage or already at the polyp stage?

Response 3: After addition of Hym-248, the polyps acquired the most Symbiodiniaceae at 72 h in Kuniya et al, 2015. We rewrote line 232-233.